# Phytate and Microbial Suspension Amendments Increased Soybean Growth and Shifted Microbial Community Structure

**DOI:** 10.3390/microorganisms9091803

**Published:** 2021-08-25

**Authors:** Bulbul Ahmed, Jean-Baptiste Floc’h, Zakaria Lahrach, Mohamed Hijri

**Affiliations:** 1Institut de Recherche en Biologie Végétale, Université de Montréal, 4101 Sherbrooke Est, Montréal, QC H1X 2B2, Canada; md.bulbul.ahmed@umontreal.ca (B.A.); jean-baptiste.floch@umontreal.ca (J.-B.F.); zakaria.lahrach.1@umontreal.ca (Z.L.); 2African Genome Center, Mohammed VI Polytechnic University (UM6P), Lot 660, Hay Moulay Rachid, Ben Guerir 43150, Morocco

**Keywords:** microbiome, phytate, soybean, phosphorus, network, MiSeq

## Abstract

Phytate represents an organic pool of phosphorus in soil that requires hydrolysis by phytase enzymes produced by microorganisms prior to its bioavailability by plants. We tested the ability of a microbial suspension made from an old growth maple forest’s undisturbed soil to mineralize phytate in a greenhouse trial on soybean plants inoculated or non-inoculated with the suspension. MiSeq Amplicon sequencing targeting bacterial 16S rRNA gene and fungal ITS was performed to assess microbial community changes following treatments. Our results showed that soybean nodulation and shoot dry weight biomass increased when phytate was applied to the nutrient-poor substrate mixture. Bacterial and fungal diversities of the root and rhizosphere biotopes were relatively resilient following inoculation by microbial suspension; however, bacterial community structure was significantly influenced. Interestingly, four arbuscular mycorrhizal fungi (AMF) were identified as indicator species, including *Glomus* sp., *Claroideoglomus etunicatum*, *Funneliformis mosseae* and an unidentified AMF taxon. We also observed that an ericoid mycorrhizal taxon *Sebacina* sp. and three *Trichoderma* spp. were among indicator species. Non-pathogenic Planctobacteria members highly dominated the bacterial community as core and hub taxa for over 80% of all bacterial datasets in root and rhizosphere biotopes. Overall, our study documented that inoculation with a microbial suspension and phytate amendment improved soybean plant growth.

## 1. Introduction

Phosphorus (P) is an essential macronutrient for all living organisms, but due to its low mobility in soil, it is a limiting factor in most agroecosystems. Plants, like other organisms, require adequate P to carry out critical functions for their growth and development [1]. Modern agriculture relies on continuous P fertilizer inputs to maintain high-yielding crop production, resulting in global P fertilizer demand increases of 2.4% per year [2]. The majority of P fertilizers used in agroecosystems are manufactured from phosphate rock (PR) which is a non-renewable resource that is inefficiently used by crop plants [3]. P fertilizer-released mineral P ions (Pi) react rapidly with binding sites in the soil environment, with plants using only about 20% of the Pi dose applied [1], and residual Pi accumulates in soil, increasing the risk of Pi-related environmental impacts [4]. That said, maintaining soil organic P (Po) is a challenging agricultural paradox.

Soil organic P (Po) is a large reservoir of P. Phytate, non-phytate phosphomonoester and phosphodiester are the three types of phosphatases that mineralize soil Po [5]. In the soil, phytate must be solubilized before phytases can cleave the orthophosphate ions available. Soil microorganisms are the main producers of phytases [6,7]; phytate hydrolysis can be aided by PGPR-based inoculants [8,9], and it is degraded in the soil by a variety of yeasts, mycorrhizal fungi, and filamentous fungi [10,11]. Hence, scientists are interested in improving plant P nutrition productivity by increasing microbial phosphatase activity. The ability of selected strains to evolve and function effectively in various soil environments is critical to the success and effectiveness of plant growth-promoting microbial inoculants for field crops. Plants play an active role in shaping soil microbiota within the soil-plant system, especially in the rhizosphere. In situ study of soil microbial interactions has proven challenging [12]. On the other hand, the ecology of soil microbial communities has been studied using within- [13] and inter-kingdom [14,15] correlation network analysis based on the cooccurrence of microorganisms. Although network analysis cannot prove the presence of taxonomic interactions, it may help to develop important hypotheses for further research on microbe-mediated soil function. Phytate hydrolysis can be spatially segregated and phosphatase activity in the root and rhizosphere soils varies, but highly efficient Po solubilizing and hydrolysing microbial strains have been isolated from forest soils [16]. We used phytate as the sole source of P. We applied a microbial suspension from soil obtained in an old growth undisturbed maple forest in Quebec’s Gault Nature Reserve (microorganisms of the suspension were not identified) as a source phytate mineralizing microorganisms, where the soil microbial assemblages are unknown. This experiment involved no other lab-grown microorganisms. The bacterial and fungal communities’ responses to microbial inoculum and phytate were represented using Illumina MiSeq amplicon sequencing. The rationale of choosing soils of maple forests to make a microbial suspension was that these soils are well known to harbour a diverse microbial community, particularly with AMF. Maple trees are able to form two types of mycorrhizal symbiosis, endo- and ectomycorrhizas [17,18], and multiple studies have documented mycorrhizal abundance in nutrient dynamics in maple forest [19,20,21]. We hypothesized that the soil of an undisturbed maple forest contains microorganisms that effectively mineralize soil phytate which would be revealed by increased crop P uptake from soil Po via phytate mineralization. Although we found no evidence of significantly increased phytate mineralization in soybean-soil systems treated with the inoculum, we did discover how microbial taxa deal with and react to phytate mineralization to make P-nutrition available to plants in underground environments. We report for the first time the evidence of Tepidisphaerales abundance in roots of a crop plant such as soybean, as well as evidence of frequent tri-party relationships involving plant, bacterium, and fungus.

## 2. Materials and Methods

### 2.1. Experimental Design, Treatments and Sampling

The study was performed in a greenhouse trial from 20 May to 24 August 2019, in a randomized complete block design with 10 blocks including four treatments with two replicates of each treatment. The average temperature during the experiment ranged from 22 to 27 °C with a photoperiod of 16/8 h. Four-liter plastic pots (7.5″ × 7.25″) were filled with substrate made from a mixture of sand, Turface and sandy loam soil (1:1:1). Turface is made by large particle size calcinated clay that absorbs excess moisture and provides, together with sand, improved water drainage and air pore space for good root growth [22]. The pots and Turface were purchased from Hydro Dionne (Montreal, QC, Canada). The sandy loam soil was collected from the 15 cm top layer of an agriculture field under organic farming at the IRDA research station in St-Bruno in Quebec (45°32′59.6″ N, 73°21′08.0″ W) on 9 May 2019. This soil had a pH 6.01, with a poor P content (0.42 ppm); its chemical properties are described in Renaut et al. 2019 [23]. The substrate mixture was made in this way to obtain a phosphate-poor medium. Seeds of Viking 2518N non-GMO soybean (*Glycine max* L.) were purchased from William Dam Seeds Ltd. (Hamilton, ON, Canada). The seeds were surface sterilized with 3% hydrogen peroxide at room temperature for 5 min and rinsed five times with sterile distilled water. The seeds were pre-germinated in water-soaked paper towel for 48 h, and then transplanted carefully in the pots. Microbial suspension (or microbial inoculum) was prepared from soil collected from the Gault Nature Reserve at Mont Saint-Hilaire. Undisturbed and old maple forests are well known as a potential source of beneficial microbes for crop plants [24]. Soil samples were taken in four distinct locations, pooled, and homogenized to form a composite sample which was then transported to the lab. Suspension was prepared by adding in deionized water, homogenized, and kept for 3 h at room temperature. Soil suspension was sieved to remove unwanted rock and plant material particles. The suspension was then transferred to a new bucket, homogenized, and set for 2 h. The supernatant was sieved through 63 µm and collected in an Erlenmeyer and 50 mL of microbial suspensions were applied in the substrate close of the plantlets after the emergence of the first two true leaves. Two treatments at four levels were performed as follows where M refers to microbial suspension and P refers to phytate: (i) M_0_P_0_ as control without microbial suspension nor phytate; (ii) M_1_P_0_ with microbial suspension without phytate; (iii) M_0_P_1_ without microbial suspension with phytate; and (iv) M_1_P_1_ with both phytate and microbial suspension. Plants were watered with tap water twice a week to keep the substrate humid. Each pot was fertilized once a week with 50 mL of full-strength Long Ashton nutrient solution without phosphate [25]. Under these conditions, plants were maintained for the entire experiment until sampling. Phytic acid sodium salt (Sigma-Aldrich, Oakville, ON, Canada) was used as a source of phytate and the dose of phytate (3.3 mg/pot) was adjusted according to previous studies [26,27]. Microbial suspension was prepared from soil collected from an old growth maple forest in the Gault Nature Reserve at Mont Saint-Hilaire, in Quebec (Appendix A).

The harvest was performed at 90 days after the experiment setup. Soybean shoots, roots and rhizosphere soil were collected per treatment and immediately transported to the laboratory and their different parameters measured. Shoots were cut at the plant collar with a clean and sterile scalpel and placed in a paper bag. Roots were separated from the soil substrate, washed with tap water, and rinsed with sterile distilled water and fragments of thin and young roots and about 5 g of rhizosphere soil (soil substrate that was attached to the roots) were collected per treatment by gently brushing the roots in a 15 mL falcon tube for DNA extraction. Shoot and root samples were placed in paper bags, kept on ice in a cooler box and immediately transported to the laboratory. The growth parameters determined were shoot fresh weight, root fresh weight and total nodules per plants. Both root and shoot samples were dried in a drying oven at 65 °C for 72 h, and their dry weights were also measured. Samples for DNA extraction were kept at 4 °C before being brought to the laboratory and preserved at −80 °C.

### 2.2. Measurement of Total Phosphorus in Plant Shoot

Total P concentration in a shoot was measured following the dry ashing protocol [28]. In brief, the top 10 cm shoots of soybean plants were taken and grinded. Grinded samples of 0.5 g were taken in a porcelain crucible, placed in a furnace muffle and ashed for 4 h at 500 °C. Once cooled, dry ashed samples were placed in a 100 mL Erlenmeyer borosilicate flask and wet with 6 mL of concentrated H_2_SO_4_ and 1 mL of HNO_3_. The solution was evaporated on a hot plate at 200 °C until the solution was clear. The solution was filtered and transferred to a volume flask and filled up with 100 mL of distilled water. The solution was stirred and immediately used to determine the total P using the molybdenum blue colorimetric method [29].

### 2.3. DNA Extraction and Amplicon Sequencing

Samples of 100 mg of roots were taken for DNA extraction using a DNeasy Plant mini kit (Qiagen, Toronto, ON, Canada) according to the manufacturer’s recommendations. Soil DNA was extracted from 250 mg rhizosphere soil samples using a DNeasy PowerSoil Pro kit (Qiagen, Toronto, ON, Canada), following the manufacturer’s suggestions. DNA was eluted in 30 μL elusion buffer and stored at −20 °C. Extracted DNAs were quantified using a NanoDrop^TM^ 2000/2000c Spectrophotometer (ThermoFisher Scientific, Ottawa, ON, Canada) and further visualized by gel electrophoresis on 1% agarose gel and a GelDoc System (BioRad, Montreal, QC, Canada). PCR amplification was performed targeting bacterial 16s rRNA and fungal ITS region and sequenced using an Illumina MiSeq at the Genome Quebec Innovation Centre (Montreal, QC, Canada). PCR amplifications of 16S rRNA were performed using forward primer CS1_341 (5′-ACACTGACGACATGGTTCTACACCTACGGGNGGCWGCAG-3′) and reverse primer CS2_806 R (5′-TACGGTAGCAGAGACTTGGTCTGACTACHVGGGTATCTAATCC-3′) [30], while fungal ITS region was amplified with CS1_ITS3_KYO2 (5′-ACACTGACGACATGGTTCTACAGATGAAGAACGYAGYRAA-3′) and CS2_ITS4 (5′-TACGGTAGCAGAGACTTGGTCTTCCTCCGCTTATTGATATGC-3′) [31]. The PCR reaction was performed in a 25 μL reaction volume composed of 1.5× Platinum^TM^ Direct PCR Universal Master Mix (ThermoFisher, Montreal, QC, Canada), 0.25 μM of each primer, 1.5× Platinum^TM^ GC Enhancer and approximately 20 ng of template DNA. Thermal cycling was performed in a Eppendorf^TM^ Mastercycler^TM^ Pro PCR system (Eppendorf, ON, Canada) with a cycling condition of activation at 94 °C for 2 min, followed by 35 cycles of denaturation at 94 °C for 15 s, annealing at 60 °C for 15 s, extension at 68 °C for 20 s and a final extension at 68 °C for 1 min with a hold at 10 °C. For each PCR run, negative PCR controls with only water were included and we did not observe any visible amplification, while amplified products were quantified and visualized onto 1% agarose gel. Amplicons were submitted for sequencing using an Illumina (San Diego, CA, USA) MiSeq sequencer through a commercial service provided by the Genome Quebec Innovation Centre (Montreal, QC, Canada). The sequence was performed with 2 × 300 bp, pair-end and demultiplexed of reads on instrument by the service provider.

### 2.4. Bioinformatics Pipeline and Processing of Data

We performed all bioinformatics, data processing and graphical analyses using R Package version R4.0.2 [32]. MiSeq reads were processed, aligned and characterized using the microbiome pipeline, DADA2, to obtain unique sequence variants known as Amplicon Sequence Variants (ASV), unlike clustering groups of similar sequences known as Operational Taxonomic Units (OTU) inferred by another pipeline QIIME [33]. In short, reads were trimmed ensuing strict quality thresholds by removing primers and poor-quality sequences with the filterAndTrim function (truncLen = c(245, 235), trimLeft = c(17, 21), maxEE = c(2,2) and truncQ = 2) (details in the complementary R scripts), followed by filtering with DADA2′s error model process using the learnErrors function. Afterward, dereplication, sample interference, merging pair-end reads and removal of chimera were performed to obtain the ASV table. The function assignTaxonomy was performed to assign taxonomy using the reference database SILVA [34] for 16S rRNA, and the UNITE database [35] for ITS. Non-fungal ASVs were removed from the fungal taxonomy dataset, reducing the number of fungal ASVs from 3230 to 2171. The identity of the ASVs of interest was manually tested using BLASTn on NCBI [36]. The bioinformatics pipeline ran remotely on multithread computer infrastructure on arcade (https://diro.umontreal.ca (accessed on 14 August 2021)).

We used the function rarecurve of the vegan package [37] to standardize the dataset to the lowest number of reads assembled for a sample by selecting randomly subsampling of the read data from each sample. The relative abundance of taxa analysed per family by exploiting the package dplyr v2.0.0 [38] in R Package version R4.0.2 [32]. For both bacterial and fungal datasets, we removed ASVs taxonomically assigned to chloroplast and mitochondria assumed that they were expected to be part of plants genome. The Alpha (α)-diversity indices Shannon and Simpson were computed using the vegan package v2.5-6 in R. The significance of treatments on α-diversity indices was tested by analysis of variance (ANOVA) and Tukey’s post hoc test was used in comparing treatments and sample types (root or rhizosphere) using the R package agricolae v1.3-3 [39]. The structure of the bacterial and fungal communities (Beta diversity) was assessed with Principal Coordinates Analysis (PCoA) based on Bray-Curtis distances using the R package vegan v2.5.6 [40]. To test the effect of treatments and sample types on community composition as a constant variable, we performed PERmutational Multivariate ANalysis Of VAriance (PERMANOVA) [41] with the function Adonis of the R package vegan v2.5-6. ASVs abundance matrix was Hellinger-transformed, and 999 permutations were used to test significance in vegan [37]. We visualized taxa abundance at the order level of community composition with the R package metacoder v0.3.4 [42]. We performed indicator species analysis using the package indicspecies v1.7.9 [43] in R4.0.2. using Šidák correction for multiple comparison in the R package ‘RVAideMemoire’ v0.9-78 [44]. The ASVs observed across microbial assemblages in all plots considered as the core microbiota. We constructed co-occurrence network to evaluate the interactions between ASVs of both root and rhizosphere microbiome using the algorithm glasso of the package SPIEC-EASI v1.0.6 [13] in R4.0.2. Subsequently, the networks were imported in Cytoscape v3.8.0 for plotting and the layout “organic” used for drawing the network [45]. Betweenness centrality and degree of connectivity score of >95% of the taxa in the network were considered in multiparticle interactions and allowed to flag the highly connected taxa as hub taxa. We tested the effects of treatments on six plant productivity measures, e.g., shoots fresh and dry weight, roots fresh and dry weight, number of nodules and total phosphorus (P) in shoots. We used two-way ANOVA with permutations to assess the statistical significance for the variables with treatments.

DADA2 was used to obtain Amplicon Sequence Variants (ASV) table and taxonomy was assigned to ASV using the reference dataset SILVA [34] for 16S rRNA, and the UNITE database [35] for ITS. The relative abundance of taxa was analysed using dplyr v2.0.0 [38] in R. The vegan package v2.5.6 [40] was used for the alpha (Shannon and Simpson) and beta diversity indices (PCoA) and performed PERmutational Multivariate ANalysis Of VAriance (PERMANOVA) [41]. Tukey’s post hoc test was performed in comparing treatments and sample types using agricolae v1.3-3 [39]. We visualized taxa abundance at the order level with metacoder v0.3.4 [42]. We performed indicator species analysis using the package indicspecies v1.7.9 [43] in R Package version R4.0.2 using Šidák correction for multiple comparison in the R package ‘RVAideMemoire’ v0.9-78 [44]. A co-occurrence network analysis was performed using the algorithm glasso of SPIEC-EASI v1.0.6 [13] and were imported in Cytoscape v3.8.0 for plotting [45].

## 3. Results

### 3.1. Soybean Biomass, Nodulation and P Nutrition Response to Treatments

We used two treatments at four levels, where M refers to microbial suspension and P refers to phytate. Treatments were labelled as M_0_P_0_ as a control with neither microbial suspension nor phytate; M_1_P_0_ as presence of microbial suspension and absence of phytate; M_0_P_1_ as presence of phytate and absence of microbial suspension; and M_1_P_1_ as a presence of both phytate and microbial suspension. Comparison of the shoot dry weight using two-way ANOVA indicated a significant (*p* ≤ 0.05) effect of phytate (M_0_P_1_) and combined treatment (M_1_P_1_) compared to control (M_0_P_0_), whereas inoculum (M_1_P_0_) was insignificant. The root dry weight was unaffected by inoculum (M_1_P_0_), phytate (M_0_P_1_) and their interactions (M_1_P_1_) (Figure 1A). Although phytate addition significantly (*p* = 0.017) increased shoot dry weight (*p* = 0.007), root dry weight (*p* = 0.037) and the number of nodules per plant (*p* = 0.017) (Table 1), the microbial inoculum had no significant influence on nodulation. Total P levels in inoculated plants (M_1_P_1_) were higher (22 mg/plant) than in control plants (M_0_P_0_) (16.78 mg/plant), but the difference was statistically insignificant (Figure 1B).

### 3.2. Bacterial and Fungal Community Structure in Different Biotopes

Raw data of Illumina MiSeq produced a total of 13,148,932 reads, with 7,763,410 reads for the bacterial 16S rRNA and 5,385,522 reads for the fungal ITS region. DADA2′s filtering, trimming and quality controlling resulted on a total of 2,896,335 reads for 16S rRNA (Appendix A) and 4,234,021 for ITS (Appendix A). Finally, we assembled forward and reverse filtered reads into 15,613 ASVs for bacteria and 2171 ASVs for fungi. We then analysed the effects of treatments on the diversity and structure of the bacterial and fungal communities in the roots and rhizosphere biotopes separately.

Alpha diversity indices of bacteria were insignificant for microbial inoculum (Shannon *p* = 0.1503 and Simpson *p* = 0.3776), phytate (Shannon *p* = 0.8643 and Simpson *p* = 0.908) and their interactions (Shannon *p* = 0.7244 and Simpson *p* = 0.5756) (Figure 1C). Similarly, fungal alpha diversity did not significantly differ for different treatments (Figure 1D). Bacterial communities clustered by niche along the first axis of the PCoA ordination. Root bacteria formed distinct clusters under M_0_P_0_, M_1_P_0_ and M_0_P_1_, whereas rhizosphere bacteria were much more scattered (Figure 1E). The clustering pattern of fungal communities showed an opposite pattern to those of bacteria. M_1_P_1_ caused fungal communities in the rhizosphere to cluster less closely than other treatments (Figure 1F). According to the PERMANOVA test, microbial inoculum had significant effect on bacteria in the root (*p* = 0.001) and rhizosphere (*p* = 0.004). Phytate significantly influenced the structure of the bacterial communities in the root biotope (*p* = 0.024) (Table 2A). PERMANOVA test was insignificant for the root fungi, but inoculum had a significant (*p* = 0.007) impact on the structure of the rhizosphere fungi (Table 2B).

### 3.3. Planctobacteria and Ascomycota Dominated Soybean Microbiota

The 15,613 bacterial ASVs were assigned to 39 phyla (Appendix A) and 196 orders (Appendix A), with Planctobacteria being the most abundant phyla in both root (Figure 2A) and rhizosphere (Figure 2C) biotopes. We chose the top 10 orders based on their high relative abundance and 8 (Tepidisphaerales, Gemmatales, Isophaerales, Pirellulales, Planctomycetales, Chthoniobacteriales, Phycisphaerales and Burkholderiales) of the 10 most abundant orders were dominant in both biotopes (Figure 2B,D). The root biotope was dominated by Tepidisphaerales (Figure 2B), while the rhizosphere biotope was dominated by Gemmatales (Figure 2D). The most representative species of soybean microbiome were identified through indicator species analysis for both bacterial (Figure 3) and fungal (Figure 4) communities. Bacterial indicator species analysis revealed 35 ASVs under inoculation treatment, with 19 ASVs enriched in the root (Figure 3A; Table 3) and 16 in the rhizosphere (Figure 3B). Thirteen ASVs were enriched under phytate treatment as indicator species, with 7 ASVs enriched in root (Figure 3C) and 6 ASVs in the rhizosphere (Figure 3D) (Table 3); however, indicator species analysis under combined inoculum and phytate (M_1_P_1_) treatment significantly (*p* ≤ 0.05) revealed BASV738 (*Tepidisphaera mucosa*) and BASV766 (*Candidatus Anammoximicrobium moscowii*) in the root and BASV1092 (*Pirellula* sp.) in the rhizosphere biotope (Appendix A).

In the fungal dataset, we identified six phyla: Ascomycota, Basidiomycota, Mucoromycota, Chytridiomycota and Blastocladiomycota, with one not assigned (NA) to any phylum (Appendix A), and 92 orders (Appendix A). Ascomycota was the most abundant phylum in both root and rhizosphere biotopes (Figure 5A). The Sordariales order dominated fungal communities both in the root and rhizosphere biotopes (Figure 5B). Both biotopes shared six (Hypocreales, Sordariales, Pleosporales, Orbiliales, Glomererellales and Pezizales) of the top 10 orders (Figure 5B). Fungal indicator species in the rhizosphere revealed 48 ASVs under inoculation treatment and nine ASVs under phytate addition. Just one fungal ASV, FASV113 (*Arthrobotrys conoides*) was found to be enriched in the root biotope under phytate treatment, while no ASV was found in the root biotope under inoculum treatment (Table 3). In the rhizosphere, four arbuscular mycorrhizal fungi (AMF), *Glomus* sp., *Claroideoglomus etunicatum*, *Funneliformis mosseae* and *Glomeromycotina* sp.; one ericoid mycorrhiza *Sebacina* sp., and three Trichoderma species, *Trichoderma aerugineum*, *T. Americanum* and *T. simmonsii*, were significantly identified as indicator species under inoculum treatment (Figure 4A); and only *Glomeromycotina* sp. was revealed under phytate treatment (Figure 4B). Indicator species analysis under M1P1 revealed FASV241 (*Sebacina* sp.) and FASV46 (*Chaetomium grande*) in the root biotope, and twelve fungal ASVs in the rhizosphere, including *Funneliformis mosseae* (Appendix A).

### 3.4. Determining Eco- and Core-Microbiota

Amplicon Sequence Variants (ASVs) ubiquitously found in every biotope (roots and rhizosphere) were defined as the eco-microbiota, and those shared between biotopes were defined as core-microbiota in soybean. One hundred ASVs were ubiquitous in all roots and 115 ASVs in each rhizosphere and they were attributed as the bacterial eco-microbiota of soybean roots and rhizosphere, respectively (Appendix A). The bacterial eco-microbiota in the root belonged to 19 genera (Appendix A), with 91 of them being Planctobacteria (Appendix A), whereas the rhizosphere eco-microbiota belonged to 21 genera (Appendix A), with 107 Planctobacteria ASVs (Appendix A). *Tepidisphaera mucosa* and *Gemmata* sp. were detected in 32 ASVs in the root and rhizosphere eco-microbiota, respectively (Appendix A). A Venn diagram identified 63 unique ASVs in the root and 78 distinct ASVs in the rhizosphere, with 37 ASVs shared by the two biotopes (Appendix A). Thirty-three of the 37 shared ASVs belonged to nine bacterial genera (*Tepidisphaera mucosa, Gemmata* sp., *Planctomyces maris, Isosphaera* sp., *Pirellula* sp., *Planctomicrobium piriforme, Lacipirellula parvula, Calycomorphotria hydatis, Algisphaera agarilytica*), while four were not assigned to any taxon (NA) (Appendix A). The shared bacterial taxa were the bacterial core-microbiota in soybean. In the fungal community, only ASV2 (*Humicola fuscoatra*) was discovered as eco-mycobiota in the root (Appendix A). Fifteen ASVs were identified as eco-mycobiota in the rhizosphere and assigned to 14 genera (Appendix A). ASV2 (*Humicola fuscoatra*) was found in both biotopes and has been considered as the core-mycobiota (Appendix A).

### 3.5. Bacteria Regulates the Connectivity of Soybean Microbiota

The interkingdom co-occurrence network in the rhizosphere was more complex (452 nodes and 2159 edges) than in the roots (285 nodes and 553 edges) (Figure 5A,B). Four bacterial ASVs (BASV6, BASV87, BASV16 and BASV58) were classified as hub taxa in the root based on their node degree and betweenness centrality and these hub taxa were Planctobacteria (Appendix A). Based on mutual putative interactions in the subnetwork of the hub taxa in the root, we found: (i) BASV6 (*Tepidisphaera mucosa*) had positive putative interactions with 13 different bacterial ASVs but negative interactions with two different bacterial ASVs (Appendix A); (ii) BASV16 (*Tepidisphaera mucosa*) interacted negatively with BASV476 but positively with nine different bacterial ASVs and a fungal ASV, FASV2 (Appendix A); (iii) BASV58 (*Gemmata* sp.) had positive putative interactions with seven different bacterial ASVs but negative interactions with four BASVs (Appendix A) and (iv) positive putative interactions with seven different bacterial ASVs and negative putative interactions with five bacterial ASVs were found in ASV87 (*Tepidisphaera mucosa*) (Appendix A). The interkingdom network in the rhizosphere identified four hub taxa, BASV175, BASV148, BASV200 and ASV311, as *Thermostilla marina*, *Chloroflexus aurantiacus*, *Zavarzinella formosa* and *Gemmata* sp., respectively (Appendix A), and their interaction pattern revealed: (i) BASV175 (*Planctomyces maris*) had positive putative interactions with 10 bacterial ASVs but putative negative interactions with six bacterial ASVs and three fungal ASVS (Appendix A); (ii) BASV200 (*Zavarzinella formosa*) had putative positive interactions with 10 bacterial ASVs and putative negative interactions with seven bacterial ASVs and a fungal ASV (FASV7) (Appendix A); (iii) BASV311 (*Gemmata* sp.) had putative positive interactions with 11 bacterial ASVs and putative negative interactions with five bacterial and a fugal ASV (FASV74) (Appendix A); (iv) BASV148 (*Chloroflexus aurantiacus*) had putative positive interactions with 16 bacterial and a fungal ASV (FASV16), and putative negative interactions with a bacterial ASV, BASV268 (Appendix A).

Meta co-occurrence patterns of hub taxa revealed a network of eight modules (Figure 6C; Appendix A): (i) Module I centered on BASV200 found in the rhizosphere was connected to Module II centered on the hub taxa BASV148 via BASV44 and BASV91; (ii) BASV35 connected Module II to Module III; (iii) Module IV centered on the root hub taxa BASV311 was linked to Module III through BASV308, and BASV577 connected Module IV and Module V; (iv) BASV14 connected Module VI and VII which are respectively centered on the hub taxa BASV87 and BASV16; (v) BASV6, the root interkingdom network’s hub taxa is linked to Module VII via BASV196, and Module VI via BASV57; (vi) BASV30 connected the Module VIII and the Module II; (vii) BASV268 and BASV132, respectively, connected Module V to Module II and the Module VIII (Figure 6C). BASV91 connecting Module I, II and VIII; and BASV308 connecting Module III, IV and VIII (Figure 6C). Overall, we identified 11 different bacterial ASVs that established interactions among eight different hub taxa to broaden interactions in soybean microbiota. As a result, these 11 ASVs and eight hub taxa, for a total of 19 ASVs, have been designated as global hub taxa (Appendix A). These 19 global hub taxa were assigned to seven genera (*Tepidisphaera mucosa*, *Gemmata* sp., *Chloroflexus aurantiacus*, *Pirellula* sp., *Ralstonia solanacearum*, *Thermostilla marina* and *Zavarzinella formosa*). Only BASV148 was a Chloroflexi member, while 18 of the 19 ASVs were Planctobacteria (Appendix A).

## 4. Discussion

### 4.1. Inoculation Tends to Influence Biomass, but Sample Types Sheltered Microbial Diversity

Our findings showed that microbial inoculation did not show any significant effects on the dry weight of shoot, dry weight root or total P in the shoots. However, phytate significantly increased the dry weight of shoot, root, and the number of nodules (Table 1). This may be due to putative phytate hydrolysis since it was the sole source of P in the procedure. Phytate bound to soil constituents can be solubilized by many microorganisms [8,16,46,47,48]; however, in comparison to phytate alone (M_0_P_1_), the combined application of inoculum and phytate (M_1_P_1_) had no significant effect on soybean P-nutrition in our study. Although many soil microbes such as *Enterobacter* sp., *Pseudomonas* sp. and *Rhizopus* sp. have been isolated and showed ability to solubilize P and tended to increase shoot fresh biomass [49]. In our study, inoculation did not increase phytate hydrolysis nor plant biomass which contradicts previous investigations where microbial inoculation significantly increased plant biomass production [50,51,52,53].

A statistically insignificant effect of microbial inoculation on phytate hydrolysis can hardly be attributed to inoculation failure. Microbial inoculum and phytate had no statistically significant impact on alpha diversity; however, according to beta diversity, inoculum significantly affected the structure of the fungal microbiota in the rhizosphere. This meant that microbial inoculum displayed a tendency for phytate hydrolysis for P-nutrition and impacted microbial community pattern. Nonetheless, the efficacy of diversity indices and species richness was tracked to see if the results of various treatments were investigated. This may be explained by the fact that multiple treatments resulted in increased microbial diversity in the rhizosphere, as well as the ability to recruit certain microbes that could be beneficial to plants [54,55]. Differences in the niche used for the analysis had an important effect on the differences observed in beta-diversity in the bacterial and fungal communities.

### 4.2. Planctobacteria and Ascomycota Predominate in the Soybean Microbiota

Planctobacteria orders, Tepidisphaerales and Gemmatales were the most abundant bacterial taxa which reported for over 80% of bacteria. Tepidisphaeraceae has recently been found abundant in soil with recurrent soybean straw returns [56] which would be coherent with a particular association between these bacterial taxa and soybean. In the rhizosphere bacterial communities associated with wild beet, Planctobacteria came second to Proteobacteria [57]. In most cases, Planctobacteria’s relative abundance ranges from 1% to 18% in soil [58], <1% to 4% in root and 5% to 18% in rhizosphere [59], and 40% more abundant in greenhouse pots than in natural soil [13]. The co-culture of *Isophaera pallida*, a Planctobacteria was found to be obligate for *Heliuthrix. uregonensis* [60] suggesting that at least some Planctobacteria may be symbiotic. We assume this is the first evidence of Planctobacteria absolute dominance in the root and rhizosphere of a healthy soybean grown in a greenhouse. The order Gemmatales was recently established by parsing genomic data from a few culturable taxa, including *Gemmata* sp. [61]. Gemmatales genera are gram negative aerobic chemotrophs and most of them are unculturable [62], meaning that they depend on biotrophic associations. As a result, Planctobacteria are largely undescribed, which explains their low taxonomic resolution and functional studies in crop plants. Planctobacteria may use organic materials in soil but not fresh plant residues, as shown by a ^13^C-labelled experiment [63], implying that they depend on associated organisms to meet their carbon requirements. Organic matter concentrations were relatively higher in the underlying soil layers in the Gault Nature Reserve, where we collected soil as a source of microbial inoculum [64]. It should be noted that the microbial association discovered in this study has never been reported and has not been used as a microbial amendment for phytate mineralization. In this study, Ascomycota was found to be the most abundant phylum. Previous studies have shown that Ascomycota is more abundant in fertilisation interactions such as carbon, nitrogen, and phosphorus [65,66]. In the rhizosphere biotope, four AMFs, an ericoid mycorrhiza and three Trichoderma were reported as fungal indicator species under inoculum treatment, and an AMF *Glomeromycotina* sp. was listed as indicator species in the root biotopes. Furthermore, an ericoid mycorrhizal fungus, *Sebacina* sp. and an AMF *Funneliformis mosseae* were reported as indicator species in roots and rhizosphere biotopes, respectively, under the treatment of M_1_P_1_. More than 80% of terrestrial plants’ roots have symbiotic relationships with AMF [20]. Despite the fact that AMF hyphae have phosphatase activity [67], only a few studies have looked into whether AMF can hydrolyze phytate and thus increase plant P uptake. Phytate was used by *Funneliformis mosseae* hyphae to transport released P to maize roots [10]. Feng and Song [68] showed that hyphae in the root-free compartment acquired P from sodium phytate and transferred it to the red clover plant, but they did not observe hyphal growth. Plants have been shown to recruit microbes from the soil that may be beneficial to them [65]. This consistency may have happened in our study since soybean plants benefitted from the mycorrhizal symbiosis.

### 4.3. Microbial Amendment Influences the Community Composition

Ten bacterial and one fungal taxa were identified as core microbiota (Figure 5D,F). We found two bacteria, *Tepidisphaera mucosa* and *Gemmata* sp., as hub taxa in the root biotope and three bacteria, *Thermostilla marina*, *Gemmata* sp., and *Zavarzinella formosa*, as hub taxa in the rhizosphere, all of which are Planctobacteria. Their wide distribution [69] and ability to degrade plant-derived polymers and exopolysaccharides formed by other bacteria has recently been documented [70]. Six Planctobacteria and a Chloroflexi were identified as global hub taxa in the co-occurrence network (Figure 5C), but no fungal taxon was included in the hub microbiota list. This indicated that bacteria had a greater impact than fungi on community assemblies and Planctobacteria may have a significant impact on multifunctionality in soybean. Planctobacteria [71] and Chloroflexi [72] were abundant in response to nitrogen and phosphorus nutrition in tomato. Overall, further research using culture-dependent approaches to isolate and characterize members of these core and global hub microbiota that may play a key role in hydrolysing phytate for P nutrition for soybean may complement our findings. Given the adaptability of global hub microbiota, we speculate that they may recruit other microbes to establish interactions and occupy multiple niches, as well as serve as a central route for outlining other microbes. The absence of fungi as hub taxa in our network study could indicate that these bacteria are highly conserved. It is also possible that abundance and distribution of bacteria in the source microbial suspension is fully dominated in their native habitat.

The soil in an undisturbed maple forest is rich in organic matter [64]. AMF were found highly associated with maple trees and serve as a nutritional hotspot [73]. Moreover, mycorrhizal hyphae and spores had previously been found in sugar maple seedlings collected in St-Hippolyte, Quebec, and Waterloo, Ontario [19,74]. This could also be a good source of beneficial bacteria and fungi [75]. We know that phytate is the most common form of organic phosphate and will be an alternate solution of the future phosphorus for agriculture; therefore, we hypothesised that identifying mycorrhizal or microbial associations involved in phytate hydrolysis could be an alternative sustainable source. Another argument for utilising this soil is that the probability of finding microbes that can hydrolyse phytate could be increased. In our study, there is no significant influence of microbial inoculum recorded in P uptake in soybean shoots. However, we found four AMF (*Glomus* sp., *Claroideoglomus etunicatum*, *Funneliformis mosseae* and *Glomeromycotina* sp.) and an ericoid mycorrhiza (*Sebacina* sp.) as indicator species in the rhizosphere of inoculated plants (Figure 3, Table 3), despite the fact that they were not reported as having phytate hydrolysis activity. There might be several potential justifications. One of the possible justifications is that Planctobacteria diversity in soil is related to soil history, including abundance and diversity, which appeared to occur highly significantly in the soil uninhabited for >45 years [76]. We prepared inoculum from an undisturbed old-growth maple forest; it was therefore rich in Planctobacteria. Microbial studies in sugar maple in Mont-Megantic, Quebec reported the abundance of Planctomycetes in the rhizosphere and Sordariomycetes (Ascomycota) as the most abundant root endophytic fungi [75]. The second possibility is that microbial community composition might be masked by the impact of non-specific changes in soil composition and growth practices in the greenhouse. The third possibility, a complementary opinion, is that Planctobacteria abundance could be a competitor for P nutrition and may be able to store P, and therefore did not respond favourably to its supply for soybean. Although the third possibility is not proved experimentally, previous reports anticipated similar circumstances for Planctobacteria-phosphorus relationships in aquatic ecosystems [77]. These possible variabilities may implicate the soil microbial composition in relation to the abundance to its origin. To the best of our knowledge, several studies have focused on the pattern of trees and microenvironmental effects on the distribution and abundance on the mountain trees of the Gault Nature Reserve from where we collected soil as the source of microbial amendments [78,79,80], but no study had been reported until recently on soil microbial diversity. Hence, we are unable to refer that such microbiological diversity and abundance might deliberate reasonable interaction benefits in soybean. Overall, our study offers a baseline understanding of microbial attributes of microbial amendments from an undisturbed old growth maple forest in a greenhouse grown crop plant.

## 5. Conclusions

The main goal of this study was to find phytate-mineralizing microorganisms using microbial inoculum from the soil of the Gault Nature Reserve in the St. Lawrence valley. The dominance of Planctobacteria in soybean root and rhizosphere communities in pot culture in the greenhouse was recorded for the first time. The diverse microbial communities discovered in this study had distinct preferences for root and rhizosphere environments. Tepidisphaerales was abundantly present in in the root biotope and Gemmatales in rhizosphere soil. The microbiota was rich in Planctobacteria, with several Planctobacteria core-taxa, and it was resilient to the introduction of a foreign microbial community. Based on the assembly of microbial communities, functionality, and interactions with plants, we speculated that the composition of microbial communities and hub taxa observed in this study should be taken into consideration when studying microbial abundance as well as their role in nutritional acquisition for the benefit of plants such as soybean. This study suggests that discovering phytate hydrolysing microbes will help us better understand how microbial amendments from an old growth maple forest respond to phytate, a phosphorus source, as well as uncover the roles of microbial taxa that have been understudied; however, further research into Planctobacteria-phytate relationships is required to enhance our understanding of how to use unknown soil microbial inoculum for crop production.

## Figures and Tables

**Figure 1 microorganisms-09-01803-f001:**
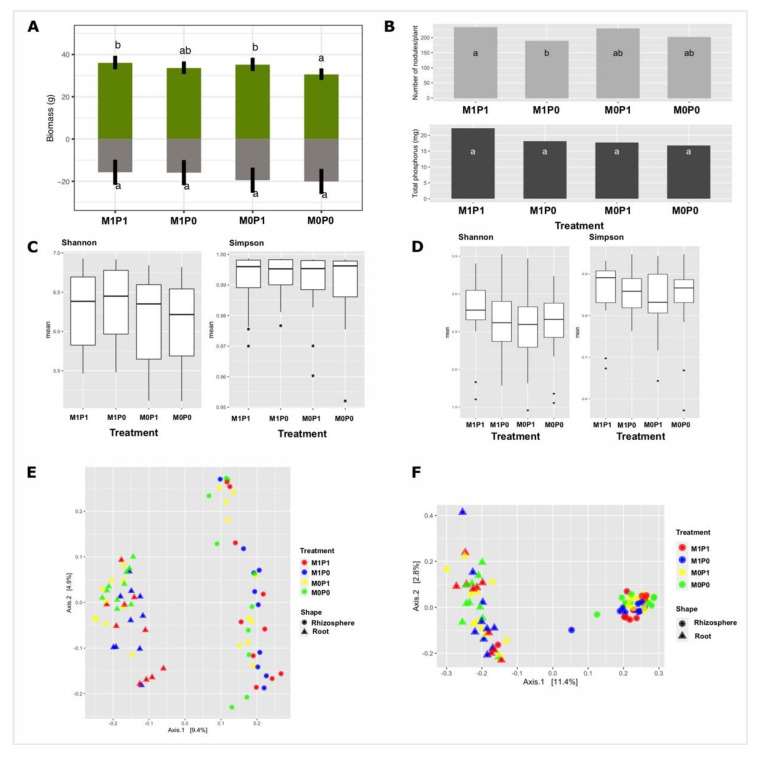
Measures of plant biomass, nodulation and total phosphorus, and microbial community structure. (**A**) Dry weight (g) of shoot and root. Green boxplots represent shoot dry weight, and grey boxplots represent root dry weight; (**B**) number of nodules per plant and total phosphorus measured in shoots. With each treatment group, means with the same letter are not significantly different by a Tukey’s range test. M_1_P_1_ = presence of both microbial inoculum and phytate; M_1_P_0_ = only microbial inoculum; M_0_P_1_ = only phytate, and M_0_P_0_ = absence of both microbial inoculum and phytate. (**C**) Shannon and Simpson diversity for bacterial microbiota; (**D**) Shannon and Simpson diversity for fungal microbiota. M_1_P_1_ = presence of both microbial inoculum and phytate; M_1_P_0_ = only microbial inoculum; M_0_P_1_ = only phytate, and M_0_P_0_ = absence of both microbial inoculum and phytate. Principal coordinates analysis (PCoA) showing the community compositions assignments of (**E**) bacterial 16S r RNA genes and (**F**) fungal ITS genes data. The variation shown in axes 1 and 2 of the ordinations is indicated in parenthesis. Circular and triangle shape represents samples from rhizosphere soil and root, respectively. Each colour represents a sample. M_1_P_1_ = microbial inocula and phytate; M_1_P_0_ = only microbial inocula; M_0_P_1_ = phytate only and M_0_P_0_ = absence of both microbial inocula and phytate.

**Figure 2 microorganisms-09-01803-f002:**
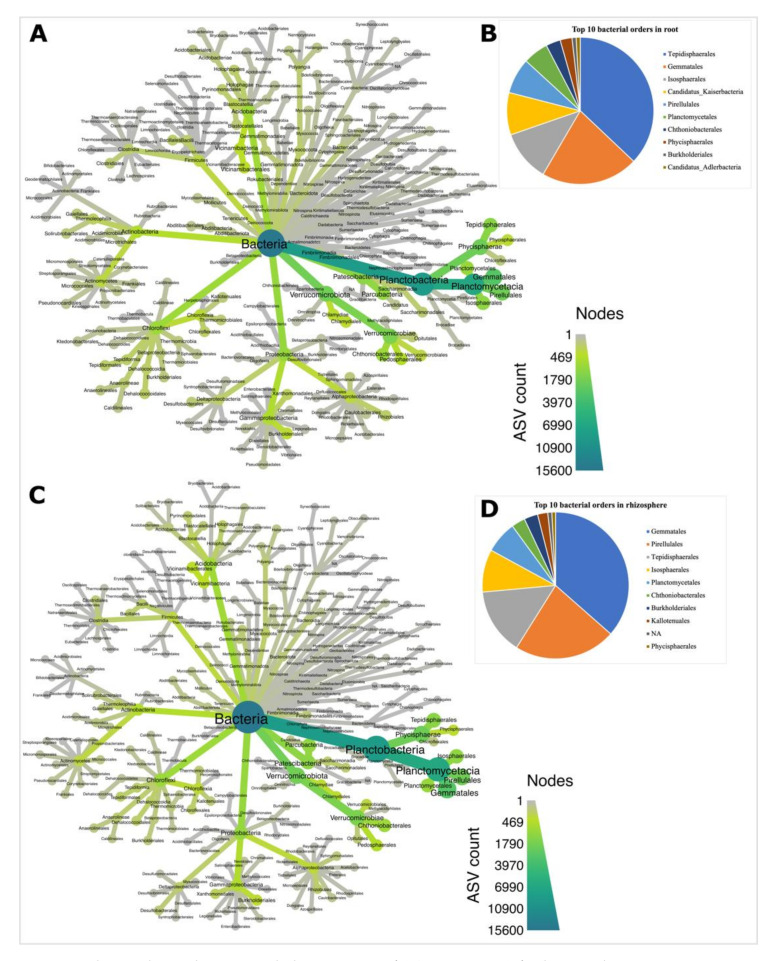
Taxonomic hierarchy and associated observations of ASVs (taxmap) for bacterial communities. Taxmap at order level in root (**A**) and relative abundance of top 10 orders in root (**B**). Taxmap at order level in rhizosphere (**C**) and relative abundance of top 10 orders in rhizosphere biotope (**D**).

**Figure 3 microorganisms-09-01803-f003:**
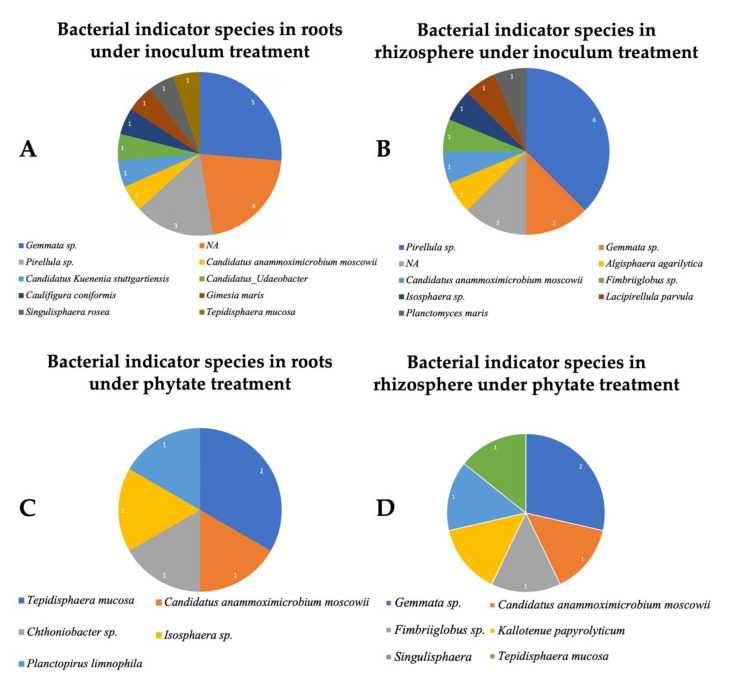
Bacterial indicator species of soybean microbiome at the genus level. The most representative species of the soybean microbiome were identified through indicator species analysis. Bacterial indicator species under inoculum treatment in root (**A**) and in rhizosphere (**B**) biotopes; and under phytate treatment in root (**C**) and in rhizosphere (**D**).

**Figure 4 microorganisms-09-01803-f004:**
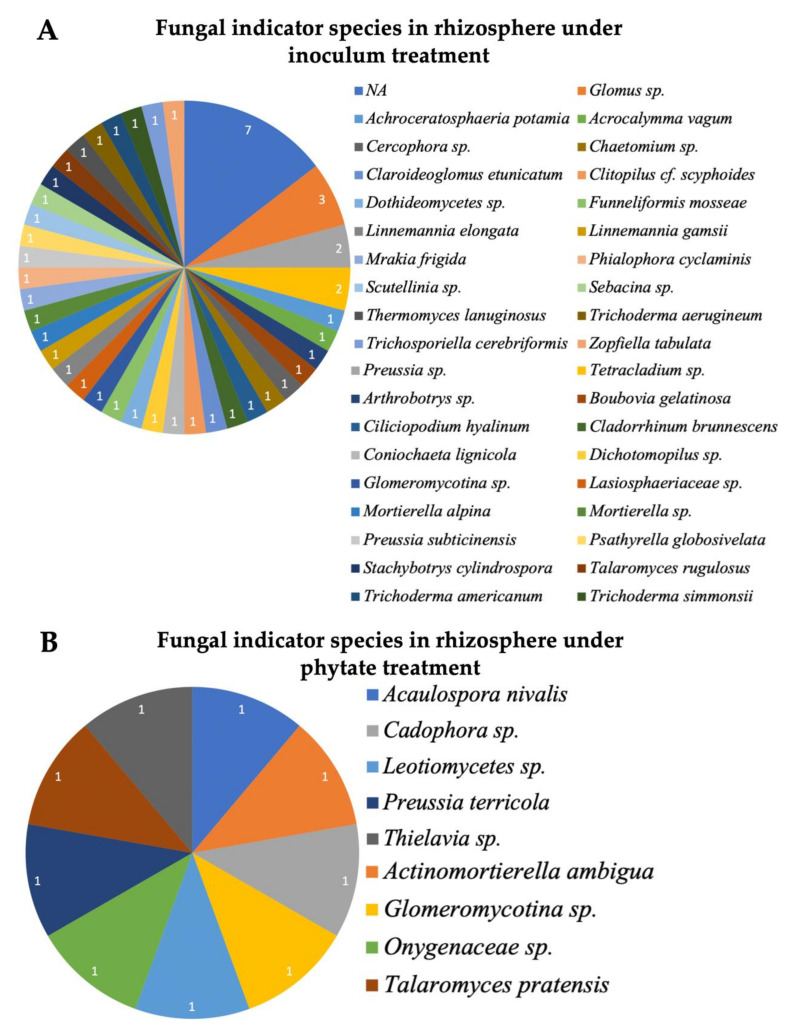
Fungal indicator species of soybean microbiome at the genus level. The most representative species of soybean microbiome were identified through indicator species analysis. Fungal indicator species in rhizosphere biotope under inoculum treatment (**A**) and under phytate treatment (**B**). The number of ASVs is indicated by the number within each taxon.

**Figure 5 microorganisms-09-01803-f005:**
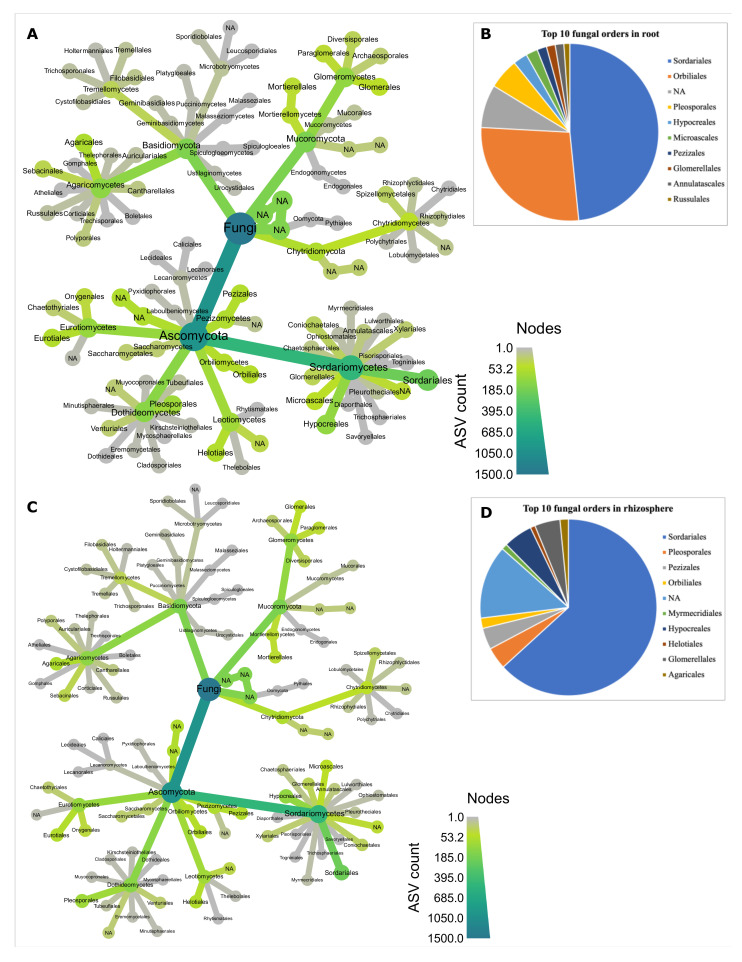
Taxonomic hierarchy (taxmap) and associated observations of ASVs (taxmap) for fungal communities. Taxmap at order level in root (**A**) and relative abundance of top 10 orders in root (**B**). Taxmap at order level in rhizosphere (**C**) and relative abundance of top 10 orders in rhizosphere biotope (**D**).

**Figure 6 microorganisms-09-01803-f006:**
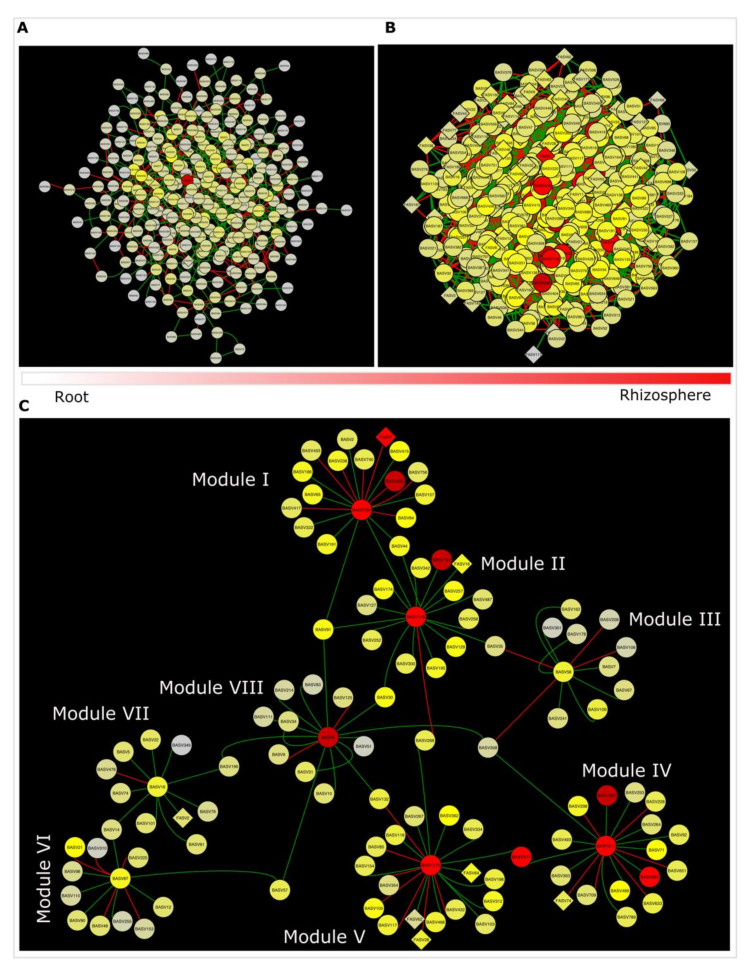
Network analysis in soybean microbiome. Inter-kingdom network in the root (**A**) and rhizosphere (**B**) biotopes. The node shapes represent bacterial (circular) and fungal (rhombus) communities. Nodes are coloured according to the relative abundance of the corresponding ASVs. The ribbon shows the relative complexity of the inter-kingdom network in root and rhizosphere biotopes. (**C**) A network build from ASVs of hub taxa and their inter-connection clustered into eight different modules. Each node represents an ASV from the microbiome. Green link represents a positive interaction and red link represents a negative interaction. Nodes are coloured according to the relative abundance of the corresponding ASVs, and node shapes denote bacterial (circular) and fungal (rhombus) ASVs. Details of the ASVs corresponding global hub taxa are in Appendix A.

**Table 1 microorganisms-09-01803-t001:** Two-way ANOVA from the effects of treatment on fresh and dry weight of shoot and root, nodule formation and total phosphorus.

Source of Variation	df	Shoot Dry Weight		Root Dry Weight		Number of Nodules		Total P	
		F	Pr(>F)	F	Pr(>F)	F	Pr(>F)	F	Pr(>F)
Inoculum	1	3.060	0.088	3.899	0.058	0.171	0.683	3.391	0.098
Phytate	1	8.018	0.007	0.037	0.847	8.525	0.017	2.901	0.122
Inoculum: Phytate	1	0.983	0.327	0.009	0.921	0.773	0.390	1.465	0.256

**Table 2 microorganisms-09-01803-t002:** Effect of microbial inoculation and phytate on the structure of the bacterial and fungal communities in root and rhizosphere according to PERMANOVA. Rhizosphere refers to rhizosphere soil which is the portion of soil adjacent and influenced by the roots, and inoculation refers to microbial suspension.

(**A**) Bacteria.						
**Variable**	**Source**	**DF**	**SumOfSqs**	**R^2^**	**F**	**Pr(>F)**
Roots	Inoculation	1	0.2347	0.04366	1.7477	0.001 ***
	Phytate	1	0.1686	0.03136	1.2555	0.024 *
	Inoculation: Phytate	1	0.1384	0.02574	1.0306	0.354
	Residual	36	4.8353	0.89924		
	Total	39	5.3771	1.00000		
Rhizosphere	Inoculation	1	0.2676	0.05702	2.2687	0.004 **
	Phytate	1	0.0878	0.01871	0.7446	0.929
	Inoculation: Phytate	1	0.0908	0.01934	0.7696	0.849
	Residual	36	4.2474	0.90490		
	Total	39	4.6937	1.00000		
(**B**) Fungi.						
**Variable**	**Source**	**DF**	**SumOfSqs**	**R^2^**	**F**	**Pr(>F)**
Roots	Inoculation	1	0.1592	0.01993	0.7662	0.751
	Phytate	1	0.1368	0.01713	0.6551	0.878
	Inoculation: Phytate	1	0.1730	0.02166	0.8283	0.653
	Residual	36	7.5181	0.94128		
	Total	39	7.9870	1.00000		
Rhizosphere	Inoculation	1	0.1502	0.04113	1.6240	0.007 **
	Phytate	1	0.0789	0.02161	0.8532	0.791
	Inoculation: Phytate	1	0.0928	0.02542	1.0035	0.428
	Residual	36	3.3296	0.91184		
	Total	39	3.6515	1.00000		

**Table 3 microorganisms-09-01803-t003:** List of bacterial and fungal ASVs selected under the inoculation and phytate addition treatments according to indicator species analysis. Significant selection at Sidak α = 0.0001, *** significant at α = 0.001, ** significant at α = 0.01, * significant at α = 0.05.

Bacteria											
Microbial Inoculation						Phytate Addition					
*Roots*			*Rhizosphere*			*Roots*			*Rhizosphere*		
Number of ASVs = 19			Number of ASVs = 16			Number of ASVs = 7			Number of ASVs = 6		
BASV5	0.0001	***	BASV590	0.0007	***	BASV1248	0.0073	**	BASV374	0.003	**
BASV8	0.0001	***	BASV861	0.0021	**	BASV107	0.0045	**	BASV761	0.0083	**
BASV114	0.0088	**	BASV569	0.0056	**	BASV1377	0.0418	*	BASV1028	0.0203	*
BASV713	0.004	**	BASV1092	0.0049	**	BASV738	0.0396	*	BASV798	0.0232	*
BASV36	0.0077	**	BASV571	0.0084	**	BASV185	0.0359	*	BASV599	0.0252	*
BASV1149	0.0163	*	BASV585	0.0086	**	BASV1514	0.0465	*	BASV619	0.0497	*
BASV264	0.018	*	BASV421	0.0183	*	BASV27	0.0426	*			
BASV590	0.0206	*	BASV537	0.0176	*						
BASV993	0.0205	*	BASV20	0.0153	*						
BASV354	0.025	*	BASV559	0.0265	*						
BASV669	0.0131	*	BASV375	0.0333	*						
BASV605	0.0188	*	BASV621	0.0357	*						
BASV250	0.0337	*	BASV587	0.03	*						
BASV381	0.0285	*	BASV882	0.0419	*						
BASV576	0.0168	*	BASV21	0.0302	*						
BASV489	0.0231	*	BASV1256	0.0359	*						
BASV1439	0.0495	*									
BASV904	0.0499	*									
BASV967	0.0247	*									
**Fungi**											
**Microbial inoculation**						**Phytate addition**					
** *Roots* **			** *Rhizosphere* **			** *Roots* **			** *Rhizosphere* **		
**Number of species = 0**			**Number of species = 48**			**Number of species = 1**			**Number of species = 9**		
			FASV410	0.002	**	FASV113	0.045	*	FASV1196	0.023	*
			FASV131	0.004	**				FASV338	0.020	*
			FASV479	0.001	***				FASV413	0.021	*
			FASV508	0.003	**				FASV798	0.034	*
			FASV400	0.009	**				FASV151	0.018	*
			FASV839	0.014	*				FASV1227	0.020	*
			FASV513	0.004	**				FASV749	0.050	*
			FASV373	0.015	*				FASV209	0.042	*
			FASV493	0.019	*				FASV366	0.029	*
			FASV503	0.007	**						
			FASV706	0.018	*						
			FASV439	0.007	**						
			FASV1286	0.015	*						
			FASV944	0.011	*						
			FASV502	0.023	*						
			FASV512	0.012	*						
			FASV551	0.027	*						
			FASV530	0.022	*						
			FASV193	0.003	**						
			FASV471	0.021	*						
			FASV885	0.016	*						
			FASV248	0.034	*						
			FASV207	0.045	*						
			FASV606	0.046	*						
			FASV328	0.038	*						
			FASV1119	0.014	*						
			FASV370	0.027	*						
			FASV261	0.035	*						
			FASV628	0.042	*						
			FASV98	0.003	**						
			FASV703	0.022	*						
			FASV790	0.006	**						
			FASV1169	0.050	*						
			FASV662	0.040	*						
			FASV1078	0.042	*						
			FASV1855	0.040	*						
			FASV702	0.045	*						
			FASV945	0.037	*						
			FASV324	0.024	*						
			FASV552	0.024	*						
			FASV465	0.041	*						
			FASV183	0.008	**						
			FASV1306	0.048	*						
			FASV299	0.040	*						
			FASV1097	0.017	*						
			FASV569	0.004	**						
			FASV108	0.029	*						
			FASV87	0.046	*						

## Data Availability

All sequences are accessible in NCBI SRA database under the accession number PRJNA720672.

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
