# Peer review of "Phytate and Microbial Suspension Amendments Increased Soybean Growth and Shifted Microbial Community Structure"

_microorganisms, 2021, doi:10.3390/microorganisms9091803_

Round 1

Reviewer 1 Report

The research work described the changes of bacterial and fungal communities associated with soybean is grown in soil amended with phytate and inoculated with PGPR. The research manuscript presents an interesting research topic and good ideas. It is well written and organized, but some analyses in the method section were not clearly described. I have made some comments which help authors to improve their manuscript.

Comment

Line 83: Why is soil mixed with sand and turface? 

Line 84: Please describe the chemical and physical properties of sandy loam soil.

Line 92: Microbial suspension? From where were microbes isolated or obtained? Are they identified? Please describe and clarify.

Line 93: “Soil samples were taken in four distinct locations”, Please explain why soil samples were taken from 4 locations, are they differ in their chemical properties?

Line 124. What about P content in soil?

Line 228: in the method section, the microbial suspension was not fully described. Would you please give more details about microbial suspension? Identified strains? Source of microbes etc.

Line 237: number of nodules? Is there specific rhizobia present in soil? Or plants inoculated with Bradyrhizobium japonicum?

Figures are very small difficult to read. Please separate figures and make sure that the text is readable.

Table 2. Please describe root and rhizosphere samples. Is “root” samples mean soil adhered from the root surface? 

If inoculation means microbial suspension, please explain in more detail about microbial inoculants.

Figure 3. Difficult to read, please separate some figures and make them readable.

Line 412: “microbial inoculum” should be explained in the method section

Line 415: “root and rhizosphere biotopes.”, Please clarify root and rhizosphere, endophytes or soil adhered from root surface? How you analyzed differences in microbial community between root and rhizosphere?

Line 425: “Many soil microbial species have been shown to be able to solubilize P”, what about microbial inoculants that used in this study? Please describe their beneficial plant traits.

Line 466: “It should be noted that the microbial association discovered in this study is most likely due to the source’s microbe availability.” Sources? Please rewrite this sentence, in this study, microbial suspension was used but no details were given about microbes, bacteria or fungi.

Author Response

We are very grateful to the reviewer#1 for taking the time to think about this work seriously and provide us with the opportunity to present a more concise paper. In view of the comments, we have modified the text of some sections to address those comments. We also increased figure 1 text and divided figure 3 in two figures to make them readable according to the reviewer's suggestion.

Responses to Reviewer’s comments point-by-point are indicated in Bold.

Line 83: Why is soil mixed with sand and turface? 

We added the following sentence to answer this question: Turface is made by large particle size of calcinated clay that absorbs excess moisture and provides together with sand improved water drainage and air pore space for good root growth [23].

Line 84: Please describe the chemical and physical properties of sandy loam soil.

We added a sentence to describe physio-chemical properties of sandy loam soil and a reference of detailed analyses in Renaut et al. 2019 [22]

Line 92: Microbial suspension? From where were microbes isolated or obtained? Are they identified? Please describe and clarify.

In line 58-61, The microbial suspension was prepared from a natural soil collected form a maple forest. Information is given in detail in Mat&Meth section. No, microbes in the suspension were not identified. This was added in L60.

Line 93: “Soil samples were taken in four distinct locations”, Please explain why soil samples were taken from 4 locations, are they differ in their chemical properties?

This was done to form a composite sample. We rephrased the sentence for clarity ‘Soil samples were taken in four distinct locations, pooled and homogenized to form a composite sample which was transported to the lab.’

Line 124. What about P content in soil?

We did not measure P content in soil after the sampling. We determined total P content in plant which is a proxy for P nutrition of plants.

Line 228: in the method section, the microbial suspension was not fully described. Would you please give more details about microbial suspension? Identified strains? Source of microbes etc.

In line 94-95, we added the detailed info with reference about microbial suspension.

Line 237: number of nodules? Is there specific rhizobia present in soil? Or plants inoculated with Bradyrhizobium japonicum?

We measured the number of nodules per plant. We did not identify any specific rhizobia taxa in the soil. We did not use any Bradyrhizobium japonicum. This was added in L61.

Figures are very small difficult to read. Please separate figures and make sure that the text is readable.

We increased the text size of Figure 1 and we divided figure 3 in two figures which are now easy to read.

Table 2. Please describe root and rhizosphere samples. Is “root” samples mean soil adhered from the root surface? 

This was done.

If inoculation means microbial suspension, please explain in more detail about microbial inoculants.

We clarified about roots in the line 118-120. For rhizosphere and inoculation we added info to the Table 2.

Figure 3. Difficult to read, please separate some figures and make them readable.

We divided Figure 3 in two figures now Fig. 3 A, B, C D and Fig. 5A, B. We also changed figure numbers accordingly.

Line 412: “microbial inoculum” should be explained in the method section

Referred to methods in the line 93.

Line 415: “root and rhizosphere biotopes.”, Please clarify root and rhizosphere, endophytes or soil adhered from root surface? How you analyzed differences in microbial community between root and rhizosphere?

We clarified about biotopes and community analysis in the line 421-424. We mentioned about root and rhizosphere in the result section (Table 2).

Line 425: “Many soil microbial species have been shown to be able to solubilize P”, what about microbial inoculants that used in this study? Please describe their beneficial plant traits.

We revised and updated.

Line 466: “It should be noted that the microbial association discovered in this study is most likely due to the source’s microbe availability.” Sources? Please rewrite this sentence, in this study, microbial suspension was used but no details were given about microbes, bacteria or fungi.

We revised and corrected the sentence.

Reviewer 2 Report

Phytate represents an organic pool of phosphorus in soil that require hydrolysis by phytase enzymes produced by microorganisms prior to its bioavailability by plants. The researchers demonstrated that soybean nodulation and shoot dry weight biomass increased when phytate was applied to the nutrient-poor substrate mixture. Bacterial and fungal diversities of the root and rhizosphere biotopes were relatively resilient following inoculation by microbial suspension; however, bacterial community structure was significantly influenced. In general terms the topic of the reviewed article is interesting. The  manuscript was prepared with care and its content contains a lot of valuable information. The work does not raise any scientific or substantive reservations.

The manuscript is well structured, the methodology is explicitly presented and the results reported are interesting. All tables and figures are clear, understandable and necessary. The references are sufficient and necessary.

The paper needs some editorial corrections.

I recommend the publication of this manuscript in the Microorganisms after minor corrections.

Author Response

We were very pleased to hear that the reviewer found the manuscript to be interesting and appropriate for Plants. We thank the reviewer for suggesting to proofread the text. We will use the service provided by the journal to edit English language.

Round 2

Reviewer 1 Report

The authors improved MS considering reviewer comments and explained each point clearly. I feel that research work is worth considering in the journal of Microorganisms.

Author Response

We thank the reviewer to reading our revision.